# Multi-Year Retrospective Analysis of Mortality and Readmissions Correlated with STOPP/START and the American Geriatric Society Beers Criteria Applied to Calgary Hospital Admissions

**DOI:** 10.3390/geriatrics8050100

**Published:** 2023-10-09

**Authors:** Roger E. Thomas, Robert Azzopardi, Mohammad Asad, Dactin Tran

**Affiliations:** 1Faculty of Medicine, University of Calgary, Calgary, AB T2N 1N4, Canada; mohammadimamhasanbin@ucalgary.ca (M.A.); dactin.tran@ucalgary.ca (D.T.); 2Oracle Canada, Mississauga, ON L5R 4H1, Canada; robert.azzopardi@oracle.com

**Keywords:** seniors, potentially inappropriate medications, potential prescribing omissions, hospital readmissions, mortality

## Abstract

**Introduction:** The goals of this retrospective cohort study of 129,443 persons admitted to Calgary acute care hospitals from 2013 to 2021 were to ascertain correlations of “potentially inappropriate medications” (PIMs), “potential prescribing omissions” (PPOs), and other risk factors with readmissions and mortality. **Methods**: Processing and analysis codes were built in Oracle Database 19c (PL/SQL), R, and Excel. **Results:** The percentage of patients dying during their hospital stay rose from 3.03% during the first admission to 7.2% during the sixth admission. The percentage of patients dying within 6 months of discharge rose from 9.4% after the first admission to 24.9% after the sixth admission. Odds ratios were adjusted for age, gender, and comorbidities, and for readmission, they were the post-admission number of medications (1.16; 1.12–1.12), STOPP PIMs (1.16; 1.15–1.16), AGS Beers PIMs (1.11; 1.11–1.11), and START omissions not corrected with a prescription (1.39; 1.35–1.42). The odds ratios for readmissions for the second to thirty-ninth admission were consistently higher if START PPOs were not corrected for the second (1.41; 1.36–1.46), third (1.41;1.35–1.48), fourth (1.35; 1.28–1.44), fifth (1.38; 1.28–1.49), sixth (1.47; 1.34–1.62), and seventh admission to thirty-ninth admission (1.23; 1.14–1.34). The odds ratios for mortality were post-admission number of medications (1.04; 1.04–1.05), STOPP PIMs (0.99; 0.96–1.00), AGS Beers PIMs (1.08; 1.07–1.08), and START omissions not corrected with a prescription (1.56; 1.50–1.63). START omissions for all admissions corrected with a prescription by a hospital physician correlated with a dramatic reduction in mortality (0.51; 0.49–0.53) within six months of discharge. This was also true for the second (0.52; 0.50–0.55), fourth (0.56; 0.52–0.61), fifth (0.63; 0.57–0.68), sixth (0.68; 0.61–0.76), and seventh admission to thirty-ninth admission (0.71; 0.65–0.78). **Conclusions:** “Potential prescribing omissions” (PPOs) consisted mostly of needed cardiac medications. These omissions occurred before the first admission of this cohort, and many persisted through their readmissions and discharges. Therefore, these omissions should be corrected in the community before admission by family physicians, in the hospital by hospital physicians, and if they continue after discharge by teams of family physicians, pharmacists, and nurses. These community teams should also meet with patients and focus on patients’ understanding of their illnesses, medications, PPOs, and ability for self-care.

## 1. Introduction

### 1.1. Background/Rationale

A key concern is whether medications assessed as “potentially inappropriate medications” (PIMs) or “potential medication omissions” (PPOs) contribute to seniors’ hospital admissions. The STOPP/START [1] and American Geriatric Society Beers [2] are the two main criteria used for assessment. Medline and Embase were searched to 20 August 2023 with the search terms patient readmission or hospitalisation and (systematic reviews or meta-analyses) to identify the frequency of readmissions of patients ≥65 and the role of medications and other risk factors.

A systematic review identified 62 studies (2 RCTs and 60 nonrandomised studies with 1,854,698 patients), and the average percentages (weighted by study size) for 30 STOPP/START studies (1,245,974 patients) receiving one or more PIMs were 42.8% for 1,242,010 community patients and 51.8% for 3964 hospitalised patients. For nineteen Beers studies (595,811 patients), the average percentages for one or more PIMs were 58.0% for 593,389 community patients and 55.5% for 2422 hospitalised patients [3].

A systematic review of 30 RCTs (n = 11,693 patients) assessed whether primary care could improve post-discharge care. In ten studies, patient information was exchanged between health professionals at and after discharge, and for the intervention group, the relative risk (RR) of 30-day hospital readmission compared with control was 0.68 (95%CI 0.56, 0.84) (16.5% vs. 30%), and for 6-month readmission, it was 0.83 (0.75, 0.92) (35% vs. 38.0%) [4].

Three prospective studies of readmissions identified increasing numbers of readmissions per patient over time. For the PAERPA study (2015–2017) of 24,500 patients >75 in the Hauts de France region, the best predictor of readmission and death was the progressive number of readmissions. The relative risk of admission steadily increased after the first admission (RR 1.8; 1.7, 1.9) to after the fifth admission (RR 3.0; 2.6, 5.5), and the risk of death also steadily increased after the first admission (RR 1.1; 1.07, 1.11) to after the fifth admission (RR 1.3; 1.1, 1.5) [5].

During the DAMAGE study of 3081 patients ≥ 75 in six acute French geriatric units, again the best predictor of admission and death was prior admissions. The relative risk of readmissions steadily increased after the first admission (RR 1.31; 95%CI 1.08, 1.60) to the fifth readmission (RR 2.66; 1.44, 5.14), and the relative risk of death increased after the first admission (RR 1.61; 1.48, 1.76) to after the fifth readmission (RR 2.01; 123, 3.32) [6].

A prospective study of 625 patients ≥70 in two Belgian general teaching hospitals found that the 30-day readmission rate was 12.3%, and the risk factors were hospitalisation in the previous three months (OR 2.21; 1.24 to 3.87), longer prior stay (OR 2.79; 1.31 to 5.67), and discharge diagnoses for respiratory disease (OR 3.49; 1.88 to 6.42) or genitourinary illness (OR 5.91; 2.24 to 14.54) [7].

There are two national retrospective cohort studies of readmissions, and again the readmission rate increased over time. In a study in New Zealand (2009–2010), 66,983 patients ≥65 had 95,318 admissions. On their first admission, 13.3% were readmitted within 30 days and 23.8% within 90 days, with 4.6% dying in hospital within 30 days and 8.5% within 90 days. On the second admission, 26.1% were readmitted within 30 days and 40.9% within 90 days, with 6.0% dying in hospital within 30 days and 15.4% within 90 days [8].

In Switzerland, the Swiss Diagnosis-Related Groups reimbursement system was adopted nationwide in 2012 to provide financial incentives to reduce the length of stay; there were, however, small increases in readmission rates. For 2,426,722 medical admissions from 2009 to 2015 (median age 70, IQR 55–81), there was a small increase in the 30-day readmission rate (14.4% pre- and 15.0% post-DRG) and a decline in in-hospital mortality (4.9% pre- and 4.6% post-DRG). However, the readmission rates pre- and post-DRG for the 398,479 patients with the five main diagnoses increased for acute myocardial infarction (from 13.7% to 22%), COPD exacerbation (from 17% to 18.2%), acute heart failure (from 14.1% to 16.1%), community-acquired pneumonia (from 10.6% to 11.3%), and pulmonary embolism (from 7.8% to 9%) [9]. Retrospective studies also identified similar high readmission rates [10,11,12,13].

Thus, there is an urgent need to identify why seniors have so many readmissions, and how the readmission rate can be decreased.

### 1.2. Objectives

To measure the relative contributions of risk factors for mortality and readmissions identified in the literature review in this retrospective database of 129,443 first admissions and 155,758 readmissions of Calgary seniors during 2013–2021;To identify cohorts at the highest risk of mortality and readmission;To identify the costs of readmissions and how much of these costs could be freed up for interventions by teams of family physicians, pharmacists, and home-visiting nurses to maintain patients as long as possible in their own homes and avoid readmissions.

## 2. Methods

### 2.1. Study Design

A retrospective cohort database study of the Province of Alberta’s Health Services DIMR database was carried out. DIMR staff anonymised admission, hospital, and discharge records (file DAD) of all individuals ≥65 years admitted to the four main acute care hospitals in Calgary with ICD-10 diagnoses and ICC procedure codes in the Province of Alberta DIMR database, and prescribing data in the Patient Information System (PIN).

### 2.2. Ethics Approval

This study was approved by the Conjoint Health Research Ethics Board (CHREB), Research Services, University of Calgary (Project REB15-2163).

### 2.3. Setting

We used a retrospective database of 129,443 first admissions of Calgary seniors and their 155,758 readmissions during 2013–2021.

### 2.4. Participants

All individuals ≥65 years admitted to the four main acute care hospitals in Calgary from 2013 to 2021 were included in this study.

### 2.5. Variables

Demographics (age, sex), medical illnesses (Medical Council of Canada codes), number of illnesses, number of medications, Charlson scores, PIMs, PPOs, corrected PPOs, readmissions, and mortality were used as variables.

### 2.6. Data Sources/Measurement

The Province of Alberta’s DIMR staff anonymised admission, hospital, and discharge records (file DAD) of all individuals ≥65 years admitted to the four main acute care hospitals in Calgary with ICD-10 diagnoses and ICC procedure codes in the Province of Alberta DIMR database, and prescribing data in the Patient Information System (PIN). The 2015 STOPP/START medications [1] and the 2019 AGS Beers criteria [2] were linked to ATC codes.

### 2.7. Bias

All output data were independently checked by two authors.

### 2.8. Study Size

All individuals ≥65 years admitted to the four main acute care hospitals in Calgary from 2013 to 2021 (129,443 first admissions and 155,758 readmissions) were included, with no exclusions.

### 2.9. Quantitative Variables

The Alberta Health Services DAD and PIN data were combined using the patient identifier and admission time windows, and any duplicate records in the provided source files were eliminated in the course of processing.

### 2.10. Statistical Methods

There is no publicly available software for applying STOPP/START or AGS Beers criteria to electronic medical records. The processing and analysis codes developed in the course of this research project were built using a combination of Oracle Database 19c (PL/SQL), R, and Excel, and statistical analyses (logistic regressions) were conducted within those databases. Patient outcomes were analysed in terms of age, sex, number of medications on admission and discharge, number of comorbidities, Charlson Index, PIMs, PPOs, and corrected PPOs.

Missing data were not replaced because the dataset was anonymised by Alberta Health Services, and we had no access to individual patient charts.

The records of all patients admitted to the four Calgary acute care hospitals were entered into a unified data system. If patients went out of province, and their records were not thereafter incorporated into the AHS database, we had no way of knowing about any out-of-province medical events.

Logistic regressions adjusted raw outcome data for age on admission, sex, and number of comorbidities.

The DAD and PIN data were combined using the patient identifier and admission time windows, and any duplicate records in the provided source files were eliminated. The strategy utilised for joining these two datasets consisted of matching prescriptions for a given patient ID by date, identifying prescribed medications one month prior to admission and 180 days post-admission, and removing any prescriptions that fell into overlapping visit date windows as well as any duplication (Figure 1). The admission window is variable and not shown to scale.

Coupled with the six-month post-discharge time window studied, the discharge supply (average 34 days, median 28) amounted to 9,342,962 post-discharge prescription records for the 129,443 patients in the study. Subsequent processing excluded from this raw count any duplicate medications along with any medications counted as part of pre-admission prescriptions. The prescriptions associated with a given visit were then categorised by type because of the wide variety of brand names and generic medications encountered over the cohort time period and also to simplify correlations with STOPP/START and AGS Beers criteria. The admission data also included diagnostic information for the visit, and this was used to calculate the criteria results for a given visit. Where required, the laboratory results for eGFR (estimated glomerular filtration rate) were included to support calculations.

The results per visit for criteria are generally binary (medication is present or not). For AGS Beers, they are ternary (values of 0, 1, or 2), with 0 meaning no medication, 1 meaning diagnosis with no recommended medication, and 2 meaning diagnosis with recommended medications.

There is no publicly available software for applying STOPP/START or AGS Beers criteria to electronic medical records. The individual criteria were coded as Oracle database functions and applied to the raw data. This design allowed for rapid processing and modularity for recalculation. The final results table was analysed using R Studio and Markdown. For calculations of odds ratios, the generalised linear model (GLM) was used to apply logistic regressions (family parameter set to binomial). The processing and analysis code developed in the course of this research project was built using a combination of Oracle Database 19 c (PL/SQL), R, and Excel.

## 3. Results

### 3.1. Summary of the Characteristics of the Patients with a First Admission in 2013–2021 Admitted to the Four Acute Care Hospitals in Calgary

In Table 1, descriptive statistics are shown for a total of 285,201 visits for 129,443 unique patients of both genders.

The number of first admissions for which data were available across all three Alberta Health Services databases (medical, surgical, and procedures; medications; and laboratory results) was 129,443 (median age 76 years). On admission, they had a median of three medications, and at discharge, they had a median of nine (Table 1).

### 3.2. Numbers of Patient Admissions to the Four Acute Care Calgary Hospitals 2013–2021

Data were obtained from three Alberta Health Services datasets, and the total numbers reported in the tables vary according to how much information could be joined across all three datasets. The initial admission in 2013–2021 was of 129,443 patients, the second was of 64,441 of those patients, the third was of 35,206, the fourth was of 20,354, the fifth was of 12,271, and the sixth was of 7577 patients. Another 14,951 patients had 7–15 admissions, and 843 had 16–25 admissions. A very small number of patients had more than 25 admissions during this period: The final 115 patients had 26–39 admissions. There were slightly more male (144,908) than female (140,293) admissions (Table 2).

### 3.3. Numbers of Admissions, Readmissions, Length of Stay, and Deaths in Hospital and within the Next Six Months

The percentage of patients who died during their first hospital admission was 3.03%, the percentage increased to 7.21% by the sixth admission, and the average for the seventh to thirty-ninth admissions was 7.54% (Table 3). Of the 129,443 first admissions, a large number (28,292; 21.9%) were readmitted within six months. The percentage of readmissions within six months rose steadily to the sixth readmission (45.2%) and then averaged 53.5% for the seventh to thirty-ninth readmissions. The percentage who died within six months of admission increased from 9.4% after the first admission to 24.9% after the sixth admission and then averaged 26.0% for the seventh to thirty-ninth admissions. The average stay rose from 9.8 days for the first admission to 13.0 days for the sixth and then averaged 11.7 days thereafter. The relative risk of readmission rose from 49.8% for a second admission after a first admission and to 61.8% for a sixth after a fifth readmission. For the smaller numbers of admissions after the seventh, the relative risk varied between 65.0% and 100%, with the numbers of readmissions becoming much smaller as the multiple readmissions progressed.

### 3.4. Principal Diagnoses for Admissions 1 to 39, and Annually for Each Year during 2013–2021

One method of visualising the large numbers of principal admitting diagnoses is to use “heat maps”, which show larger numbers of patients in hot colours (e.g., red) and smaller numbers in cooler colours (pale yellow and green). The principal admitting diagnoses are shown in Figure 2 and Figure 3 as heat maps, and the numerical data are also shown in the text below.

The most frequent principal admitting diagnoses (Figure 2) were for cardiac problems (34,277): heart failure (12,159); arrhythmias without cardiac catheterisation (4031); percutaneous coronary intervention with MI/shock/arrest/heart failure (3706); pacemaker implantation (3555); myocardial infarction/shock/arrest without cardiac catheterisation (2608); other cardiac disorders (2326); cardiac valve replacement (2206); percutaneous coronary intervention without MI/shock/arrest/heart failure (1932); and coronary artery bypass without cardiac catheterisation, without MI/shock/arrest, with/without pump (1724).

The second most frequent principal admitting diagnoses were orthopaedic procedures (25,610): knee replacement (10,139); hip replacement (5924); fixation/repair hip/femur (3754); spinal vertebrae intervention (3106); and hip replacement with trauma/complication of treatment (2687).

The third most frequent principal admitting diagnoses were infections (22,364): viral/unspecified pneumonia (6019); lower urinary tract infection (5334); other/unspecified septicemia (3276); enteritis (3143); severe enteritis (2402); and aspiration pneumonia (2190).

The fourth most frequent principal admitting diagnoses were related to pulmonary diseases (15,637): chronic obstructive pulmonary disease (COPD) (10,995; pulmonary embolism (1944); and malignant neoplasms of the respiratory system (1698). The fifth most frequent principal admitting diagnoses were associated with the central nervous system (13,308): ischemic event central nervous system (4884); organic mental disorder (4633); and dementia (3791). The sixth most frequent principal admitting diagnoses were related to renal diseases (8890): minor intervention on the upper urinary tract (3572); renal failure (2604); nonmajor intervention on the lower renal tract (2409), and disorder fluid/electrolyte balance (2305). The seventh most frequent principal admitting diagnoses were gastrointestinal diseases (6505): gastrointestinal haemorrhage (2962); symptom/sign of digestive problems (1848); and laparoscopic cholecystectomy (1695). There were 5897 operations for partial excision/destruction of the prostate.

This pattern of serious illness was also seen in each of the admissions in each of the years 2013–2021. Figure 2 lists the top 20 of the 40 most frequent diagnoses. The Medical Council of Canada (MCC) diagnoses (Figure 3) with the darker red colour highlight the anatomic systems with the most numerous primary admissions and show the persistence of admissions with these organ system problems.

### 3.5. The Number of Medications Pre-Admission and on Discharge for All Admissions

At their first admission, the 129,443 patients took a median of three medications, and on discharge, they took a median of nine medications. The maximum number of medications increased from 24 on admission to 45 on discharge (Table 1, Figure 4).

Of the top ten most frequently prescribed medications, seven were for cardiac conditions: beta-blockers (3,987,449 prescriptions), statins (2,706,992 prescriptions), calcium channel blockers (2,343,022 prescriptions), ACE inhibitors (1,786,653 prescriptions), antiplatelet agents (1,724,720 prescriptions), and ARBs (1,363,172 prescriptions) (Appendix A).

### 3.6. Costs of Admissions during 2013–2021 Measured Using Resource Intensity Weighting

The costs for the first to nineteenth admissions are shown. (Data are not shown beyond the nineteenth readmission because of the small numbers.) The Canadian Institute of Health Information (CIHI) designed the resource intensity weighting (RIW) method to estimate the costs of individual patient hospital admissions across Canada, with separate estimates for each province. The “cost per weighted case” (CPWC) 2022 measures a ratio for each province and for the whole of Canada. This ratio is a hospital’s total acute inpatient care expenses compared with the number of acute inpatient weighted cases. The weighted cases used were CMG+ 2022 (CIHI’s most recent case-mix grouping methodology) [14,15]. Payments to physicians are in a separate confidential file we could not access. Resource intensity weights measure the intensity of resource use (i.e., relative cost) for the diagnostic, surgical, procedure, and medical care of an individual. RIWs are assigned according to the case-mix group to which an individual is assigned and also include the patient’s age, health status, and discharge status. For micro-costing, we used the CIHI complexity overlay (CMG Plx™). We computed the RIW data, and they were skewed for each year during 2013–2021 (due to some very expensive admissions). However, because this is a very large dataset comprising nine years of financial data, it is expected that the skewness will be a persistent feature of the RIW data, and the RIW methodology has been tested by CIHI so that they form the most reliable available data on costs. The average cost of a hospital stay in Alberta in 2020–2021 was CAD 9149, and for Canada, it was CAD 7619. In the Calgary dataset 2013–2021, the estimated RIW of a senior’s admission ranged from 1.527 to 11.727 times more expensive than the average cost of admission (Figure 5). The heat map shows the consistently high costs (red colour) occurring in admissions for ear, nose, mouth, and throat diseases; dementia; rehabilitation; burns; circulatory system; cancer; and the respiratory system. RIWs tended to be very high when complex cancer surgery and reconstruction were involved (Figure 5). Using the RIW cost data available for the 285,201 patients for whom we had data from all three AHS databases, we extrapolated the costs for the full sample of 295,708 patients. Using Alberta costs, the total cost of all seniors’ admissions to the four acute care hospitals in Calgary during 2013–2021 was CAD 7,477,391,068, and if average costs in Canada were considered, the total cost was CAD 6,571,010,108. Using Alberta costs, the cost of the second to thirty-ninth admissions during 2013–2021 was CAD 2,762,343,130. If 10% could be saved from the second to thirty-ninth admissions, that would provide CAD 276,234,313 for teams of family physicians, home-visiting nurses, pharmacists, and other professionals to keep senior patients at home and avoid readmissions. If 20% could be saved, then CAD 552,468,626 would be available. Seniors’ admissions include many surgeries, and there would be limited opportunity for cost savings from surgeries.

According to the Canadian Institute of Health Information, the cost of a standard hospital stay (also referred to as “cost per weighted case” (CPWC)) 2022 measures a ratio for each province and for the whole of Canada. This ratio is a hospital’s total acute inpatient care expenses compared with the number of acute inpatient weighted cases. The weighted cases used were CMG+ 2022 (CIHI’s most recent case-mix grouping methodology). For micro-costing, we used the complexity overlay (CMG Plx™) methodology [14,15].

### 3.7. Correlations with Readmissions and Mortality 2013–2021

For all admissions (Table 4), the unadjusted odds ratios for readmission were markedly elevated for START PPOs not prescribed (1.58; 1.55–1.61), STOPP PIMs (1.17; 1.17–1.17), and to a lesser extent AGS Beers PIMs (1.11; 1.11–1.12).

The most adverse unadjusted odds ratios for mortality within six months were for START PPO medications, which were needed but not prescribed (2.16; 2.11–2.21). Patients with higher Charlson scores also had a high odds ratio of mortality (1.43; 1.42–1.43). Patients assessed as having any resource intensity weighting above the average for all admissions also had higher odds ratios of mortality (ranging between 1.39 for RIW 1–2 and 2.86 for RIW > 9). The number of comorbidities increased the odds ratio of mortality (1.19; 1.18–1.19) and also the number of medications assessed after admission (1.19; 1.18–1.19). The only unadjusted risk factor that had ORs including unity was START PPOs, which were correctly prescribed and the odds ratio of mortality was 0.99 (0.97–1.01).

When odds ratios were corrected for age, gender, and comorbidities, the risk of mortality for STOPP PPOs not corrected remained elevated (1.56; 1.50–1.63), but when a prescription was given, the ORs were markedly reversed (0.51; 0.49–0.53), and this was the only factor that provided improvement (and a major improvement) in the odds ratios of mortality.

For the first admission (Appendix A), the corrected odds ratio for readmission was elevated for START PPOs not prescribed (1.39; 1.35–1.42), STOPP PIMs (1.16; 1.15–1.16), AGS Beers PIMS (1.11; 1.11–1.11), and number of medications (1.12; 1.12–1.12). Odds ratios were elevated for correctly prescribed STOPP PIMs (1.26; 1.23–1.30). Interpretations are that these patients received correct prescriptions, but their illnesses needed time to improve, or that they received increased supervision, resulting in readmissions.

The ORs (adjusted for age at admission, gender, and comorbidities) for mortality within six months were elevated for START PPOs not corrected with a prescription (1.56; 1.50–1.63), but mortality was markedly reduced if the PPOs were corrected by prescribing for the patient (0.51; 0.49–0.53).

The next result that was of key interest in this research project was assessing whether the odds ratios of further admissions and mortality improved or worsened during subsequent admissions. The same pattern of a marked reduction in mortality if PPOs were correctly prescribed persisted in all subsequent admissions (Appendix A).

However, the odds ratios of readmission were consistently higher if START PPOs were corrected: second admission (1.36; 1.31–1.41); third admission (1.42; 1.36–1.49); fourth admission (1.50; 1.41–1.59); fifth admission (1.49; 1.38–1.60); sixth admission (1.64; 1.49–1.81), and seventh to thirty-ninth admission (1.53; 1.41–1.65). It will be important to examine each patient’s illness and progress to identify the reasons for this apparently anomalous result. The possibilities are that the patient has other illnesses that cause admissions; prescribing to correct PPOs identified illnesses and their amelioration is still underway; the patients received increased monitoring and were more likely to be admitted for follow-up care; or the patient, carer, and primary care team were concerned about deterioration from other illnesses and sought hospital care.

### 3.8. Destination after Discharge from the Four Calgary Acute Care Hospitals during 2013–2021

After their admissions, only 44% of patients returned directly to their homes or to a lodge (residences in Canada in which patients are expected to be able to look after themselves with minimal supervision) (Figure 6). An important focus of this project was identifying the experiences of the Calgary seniors during their multiple admissions. Increasing the number of direct discharges to homes with continued long-term residence in their own homes supported by a team of family physicians, pharmacists, and home-visiting nurses would be an outstanding benefit.

## 4. Discussion

### 4.1. Key Results

There was a dramatic decrease in mortality across all admissions when PPOs were corrected with a prescription (OR 0.51; 95%CI 0.48–0.53; *p* < 0.001), but a marked increase in mortality was observed if PPOs were not corrected with a prescription (OR 1.56; 1.50–1.63; *p* < 0.001). There was an increase in mortality if patients received an AGS Beers PIM (OR 1.08; 1.07–1.08; *p* < 0.001) but not a STOPP PIM (OR 0.99; 0.96–1.00; *p* < 0.001) (all data corrected for age, gender, and comorbidities).

There was an increase in readmissions if START PPOs were not corrected (OR 1.39; 1.35–1.42; *p* < 0.001) but also if they were corrected (OR 1.26; 1.23–1.30; *p* < 0.001), and the possible reasons for this unexpected finding need to be investigated. The possibilities are that the patient acquired other illnesses, prescribing to correct PPOs was underway but had not ameliorated the illnesses, or patients received increased monitoring during readmissions. STOPP PIMs also correlated with readmissions (OR 1.16; 1.16–1.16; *p* < 0.001), as did AGS Beers PIMs (1.11; 1.11–1.11; *p* < 0.001).

### 4.2. Admission Costs

When hospital costs for the province of Alberta were applied to the Calgary sample, the total cost of all seniors’ admissions to the four acute care hospitals in Calgary during 2013–2021 was CAD 7,477,391,068, and when the average Canadian costs were utilised, the costs were CAD 6,571,010,108. For the second to thirty-ninth admissions during 2013–2021, the Alberta cost was CAD 2,762,343,130. If 10% were saved by reducing the numbers of 2nd to 39th admissions, this would provide CAD 276,234,313 for home care, and a 20% saving would provide CAD 552,468,626.

### 4.3. Number of Medication on Admission and at Discharge

On admission, 50,000 patients had only one medication, and 8000 had eight medications, whereas at discharge, 6000 had one medication, and 24,000 had eight medications. The supplemental tables show that, over the 2013–2021 period, there was no improvement in reducing the number of PIMs or PPOs without a prescription. This is a key area for interventions by teams of family physicians and pharmacists.

### 4.4. Previous Studies of Interventions to Reduce PIMs, PPOs, and Admission Costs

Two key systematic reviews emphasised the importance of comprehensive patient-centred rather than system-centred interventions. A systematic review and meta-analysis of 42 RCTs of the complexity of interventions to prevent readmissions of those ≥65 rated most studies on the Cochrane Risk of Bias tool at low risk, with the most frequent problem being no reliable method of dealing with missing data. The relative risk of 30-day readmission was lower in the intervention arms (0.82; 0.73, 0.91) and was lower when the intervention augmented patient self-care capacity (RR 0.68; 0.53, 0.86) compared with when it did not (RR 0.88; 0.80, 0.97); had five unique components (RR 0.63; 0.53, 0.76) compared with when it did not (RR 0.91; 0.81, 1.01); and when at least two individuals were involved in intervention delivery (RR 0.69; 0.57, 0.84) compared with when they were not (0.87; 0.66. 0.98). Compared with Category 1 interventions (which provide no comprehensive support), Category 3 interventions provide a consistent and comprehensive strategy that emphasises the assessment and treatment of factors of patient context and self-care (caregiver contributions, functional status, impact of comorbidities, patient and caregiver goals for care, and potential for self-management) and socioeconomic factors. Studies with Category 3 interventions had a lower relative risk of admission of 0.63 (0.43, 0.91) [16].

A systematic review of 15 systematic reviews of integrated care assessed the average AMSTAR 1 score of the reviews as 7 and identified whether the 219 included studies incorporated all the elements in the WHO Integrated Care for Older People approach (WHO 2016). All the interventions included case management and multidisciplinary planning and/or care delivery and multiple care providers (nurses in 12 studies, physiotherapists in 10 studies, family physicians/GPs in 9 studies, and social workers in 9 studies). However, the outcomes in most reviews were service- or system-centred and not focused on patient care experiences, whereas the key WHO focus is for interventions to be patient-reported outcome measures (PROMs) and patient-reported experiences (PREMs) [17].

Three systematic reviews of specific interventions (physician–pharmacist collaboration, multidisciplinary teams, and pharmacist counselling) assessed the evidence as low quality, and thus conclusions about these interventions cannot be drawn. A systematic review of physician and pharmacist collaboration in primary care to reduce readmissions of adult patients included 16 RCTs. The Cochrane Risk of Bias tool assessed 16/16 studies as low risk for randomisation, 11/16 with allocation concealment, 0/16 with the blinding of participants and personnel, 15/15 with the blinding of outcome assessors, 15/16 with complete outcome data, and 16/16 with no selective reporting, but the GRADE study quality was low. Despite face-to-face communication between pharmacists and primary care physicians in all studies, there was no significant effect on readmissions (RR 0.92; 0.80–1.06) and low certainty of evidence [18]. A systematic review of 11 RCTs (n = 7496 ≥ 65) of care in acute geriatric units on the Cochrane Risk of Bias tool assessed four at high risk of bias for randomisation and two for allocation concealment, and only five performed an intention-to-treat analysis. The multidisciplinary team included a geriatrician and/or primary care physician and a nurse with geriatric training, and there was only a small difference in the relative risk of living at home after three months (RR 1.06; 0.99, 1.13) [19].

A systematic review identified 62 RCTs of pharmacists counselling patients, and for 29 RCTs of pharmacists counselling patients to adhere to their prescribed medications, adherence increased (RR 1.30; 1.19, 1.43), but the GRADE assessment of quality was low to very low. For 11 RCTs that assessed effects on 30-day readmission rates, the RR was 0.76 (0.58, 0.99) [20]. Another systematic review of 18 RCTs of pharmacist medication reviews rated 10/18 as high risk on the Cochrane Risk of Bias tool and readmission rates did not differ between the experimental and control groups (RR 0.97; 0.89, 1.05) [21].

Two small studies identified that polypharmacy and PIMs predicted readmissions. A study of 647 patients in acute geriatric wards in southern Italy in 2013 found that polypharmacy predicted 3-month readmissions (OR 2.72; 1.48–4.99) but STOPP (1.60; 0.85, 3.01) and AGS Beers criteria (0.85; 0.46–1.56) did not [22]. In a study of 259 patients discharged from a general medicine unit in Scotland during a 41.5-month follow-up period, 50% died, and a PIM was associated with three or more readmissions (OR 2.43; 1.19–4.48) and with mortality (OR 2.51; 1.20, 5.28), whereas a PPO was associated with mortality (OR 1.88; 1.09, 3.27) [23].

Thus, the key concepts in planning interventions to reduce readmissions are presented in Leppin’s [16] systematic review, which found that reductions in readmission rates were most likely if interventions improved self-care capacity, had five or more unique components, and involved two or more intervenors.

### 4.5. Strengths

This database is the largest of first admissions (129,443) and readmissions (155,758) over the longest period of nine years (2013–2021) in the literature.

### 4.6. Weaknesses

We were not able to obtain the costs that physicians billed for hospital care as this was in a separate confidential file. We did not have access to individual patient files and were unable to discover why physicians corrected some PPOs with a prescription and other PPOs not or the reasons why additional medications were started during readmissions (some were PIMs). We were unable to identify adverse drug reactions because, during the time of the study, there was no specific category for ADRs in the hospital records. We would have preferred to use Cox proportional hazards models to accurately assess survival, but the nonproportional hazard assumption was not met, so we used logistic regression.

### 4.7. Generalisability

The study is generalisable to the Calgary population as a whole because the numbers of patients by age group and gender admitted to the four Calgary acute care hospitals are representative of the Calgary population as measured by the 2022 Canadian Census (Appendix A).

The study population is also similar to the five-year age groupings of Alberta, Canada, and the US and could be generalised to those jurisdictions with appropriate caution for differences in ethnic composition, access to medical care, and prescribed medications.

However, generalisability could be reduced for populations with a substantially different ethnic composition. Twelve key P450 isoforms metabolise more than 70% of medications. Many medications also either inhibit or enhance metabolic rates in these isoforms, thus affecting how other medications are metabolised. Ethnic differences within populations are particularly important because of the major differences between ethnic groups in their P450 enzymes; the alleles they inherit from each parent; and how these affect drug metabolism, drug–drug, and drug–gene interactions. For example, two thirds of anti-depressants are metabolised by two isoforms, CYP2C19 and CYP2D6. Patients with the CYP2C19 Null/Null genotype have 0% metabolic activity for medications that are metabolised by the CYP2C19 P450 isoform, whereas those with CYP2C 19*17/*17 have two ultrarapid metabolising alleles, and they metabolise at 120% of the normal rate; for the CYP2D6WtX3 genotype (which has two ultrarapid alleles), the metabolic rate is 150% [24,25].

## Figures and Tables

**Figure 1 geriatrics-08-00100-f001:**
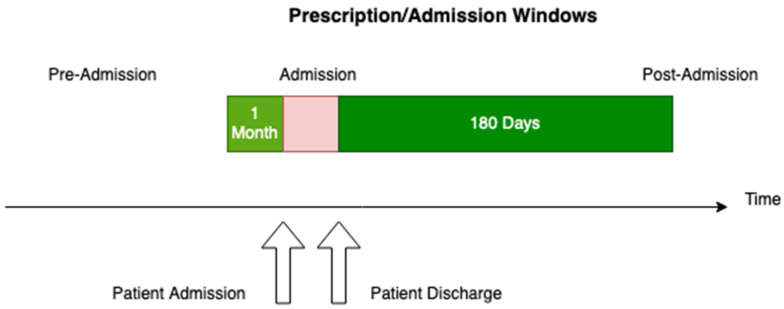
Prescription/admission time windows used for analysis.

**Figure 2 geriatrics-08-00100-f002:**
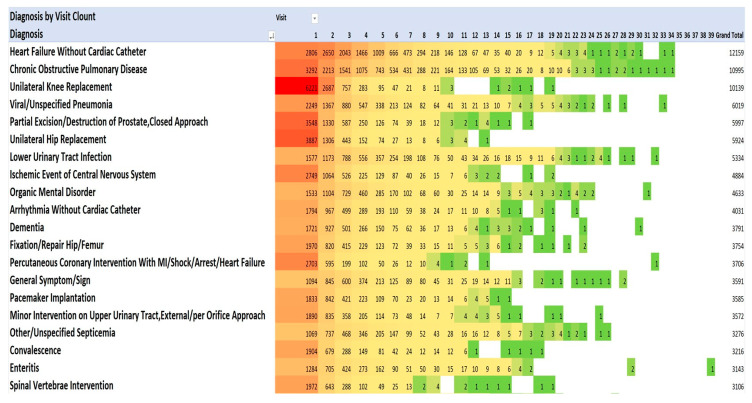
Diagnoses for each admission by number of admissions (Top 20). Red signifies highest value, green the lowest, yellow: 50th percentile.

**Figure 3 geriatrics-08-00100-f003:**
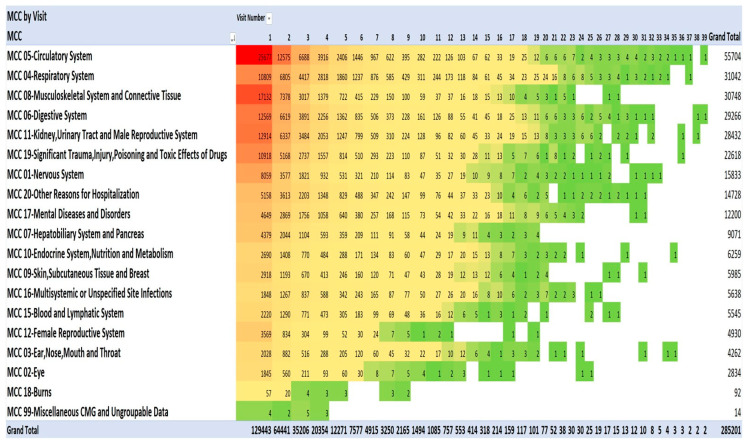
Diagnoses using Medical Council of Canada diagnostic categories. Red signifies highest value, green the lowest, yellow: 50th percentile.

**Figure 4 geriatrics-08-00100-f004:**
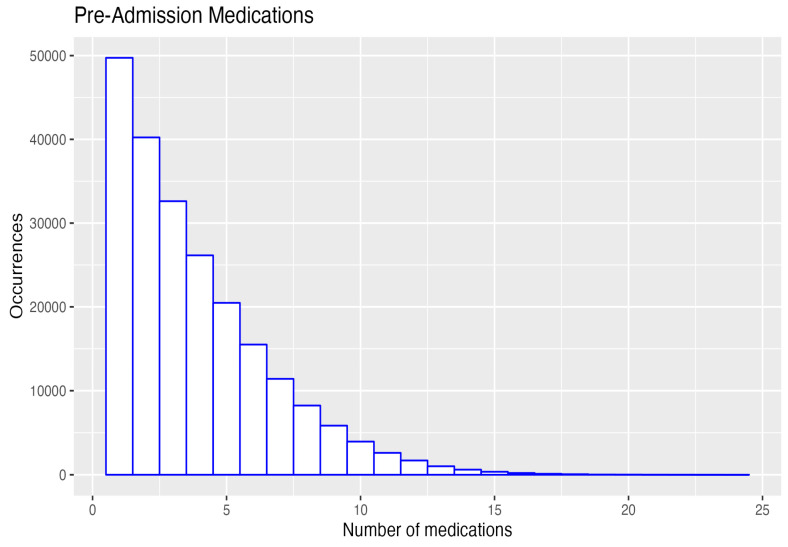
Pre- and post-admission number of medications 4.6. Class and number of medications prescribed: all admissions 2013–2021.

**Figure 5 geriatrics-08-00100-f005:**
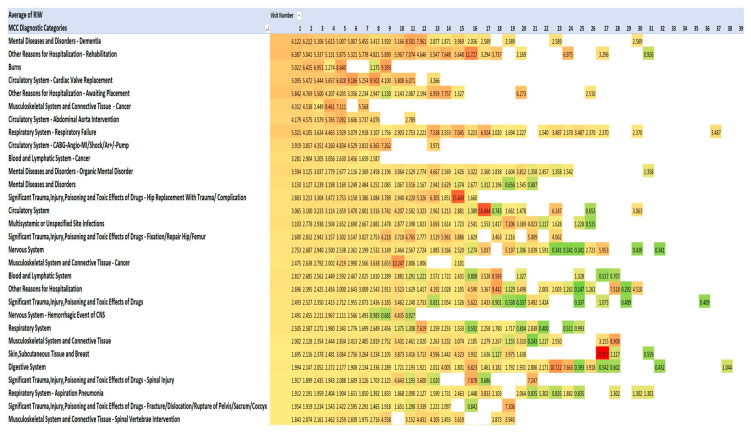
Costs of admissions measured using resource intensity weighting (RIW)—top 30. Red signifies highest value, green the lowest, yellow: 50th percentile.

**Figure 6 geriatrics-08-00100-f006:**
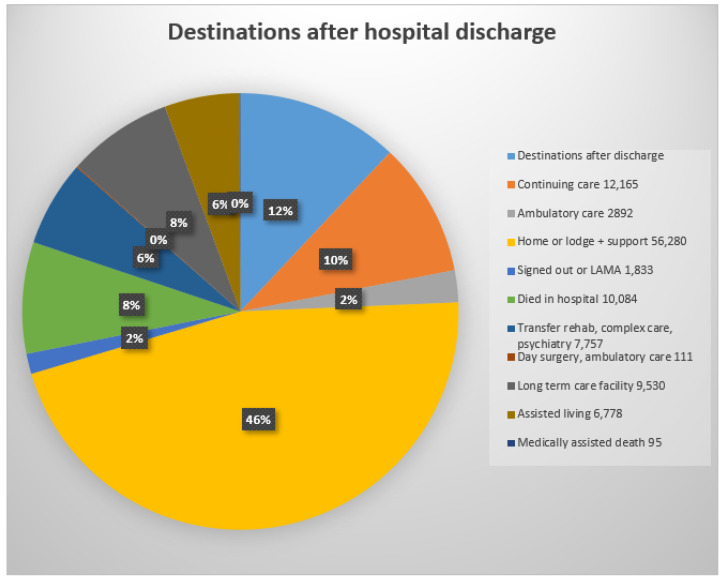
Destinations after discharge: acute care = transfer to acute care ward in current or another acute care hospital; continuing care = care in a supportive living institution or supportive care at home or in a senior’s lodge; ambulatory care = care as an outpatient including diagnostic and treatment; lodge = seniors’ home with food services and minimal supervision; sign-out = patient signed out of hospital: LAMA = left against medical advice; transfer rehab, complex care, or psychiatry = transfer to specialised ward or institution; long-term care facility = specialised long term facility or nursing home; assisted Living = residence with care provided according to patient’s needs (0% = 6 patients admitted from prison and discharged to prison; software reduced 6% to 0%).

**Table 1 geriatrics-08-00100-t001:** Summary characteristics for patients on first admission to four acute care Calgary hospitals, 2013–2021.

	Both Genders	Female	Male
Number of patients (%)	129,443 (100%)	64,621 (49.92%)	64,822 (50.08%)
Age group at first admission			
65–69	45,244 (27.39%)	20,500 (15.84%)	24,744 (19.12%)
70–74	26,042 (20.09)	12,409 (9.59%)	13,633 (10.53%)
75–79	21,526 (17.78%)	10,750 (8.30%)	10,766 (8.32%)
80–84	18,125 (16.11%)	9647 (7.45%)	8478 (6.55%)
85–89	12,138 (11.93%)	7027 (5.43%)	5111 (3.95%)
90+	6368 (6.68%)	4288 (3.31%)	2080 (1.61%)
	Total Visits 285,201		
For entire dataset			
Median age	76	77	75
IQR (age)	13	14	13
Maximum (age)	108	108	106
Medicines upon admission			
Median	3	3	3
IQR	3	3	3
Maximum	24	23	24
Medicines upon discharge			
Median	9	9	9
IQR	7	7	6
Maximum	45	45	43
Visits with readmission within 6 months	82,524 (28.93%)	38,953 (47.20%)	43,571 (52.80%)
Visits with mortality within 6 months	41,479 (32.04%)	19,554 (15.11%)	21,925 (16.94%)

Out of the population of 129,443 individuals, a total of 47,886 (36.99%) patients were documented as having died over the course of the study period 2013–2021.

**Table 2 geriatrics-08-00100-t002:** Number of admissions per patient in 2013–2021, four acute care Calgary hospitals.

Admission No	Number of Patients			Admission No	Number of Patients			Admission No	Number of Patients		
	Female	Male	Total		Female	Male	Total		Female	Male	Total
1	64,622	64,821	129,443	14	214	200	414	27	8	9	17
2	31,714	32,727	64,441	15	166	152	318	28	7	8	15
3	17,154	18,052	35,206	16	106	108	214	29	7	8	13
4	9809	10,545	20,354	17	80	79	159	30	7	5	12
5	5820	6451	12,271	18	56	61	117	31	6	4	10
6	3540	4037	7577	19	49	52	101	32	5	3	8
7	2338	2577	4915	20	33	44	77	33	2	3	5
8	1524	1726	3250	21	21	31	52	34	2	2	4
9	1034	1131	2165	22	16	22	38	35	2	1	3
10	717	777	1494	23	13	17	30	36	2	1	3
11	534	551	1085	24	13	17	30	37	1	1	2
12	379	378	757	25	11	14	25	38	1	1	2
13	271	282	553	26	8	11	19	39	1	1	2
TOTAL									140,293	144,908	285,201

**Table 3 geriatrics-08-00100-t003:** Numbers of first admissions, readmissions, deaths in hospital, length of stay, deaths in next 6 months, and risk of readmission.

Admission	Patients	Mort Ality in Hospital	% Died in Hospital	Readmitted within 6 Months	% Readmitted within 6 Months	Died within 6 Months	% Died within 6 Months	Average Stay (Days)	Relative Risk of Readmission
1	129,443	3919	3.03%	28,292	21.9%	12,144	9.4%	9.8	
2	64,441	2947	4.57%	18,157	28.2%	9427	14.6%	11.6	49.8%
3	35,206	2049	5.82%	11,585	32.9%	6500	18.5%	12.6	54.6%
4	20,354	1414	6.45%	75,76	37.2%	4450	21.9%	13.2	57.8%
5	12,271	885	7.21%	4982	40.6%	2948	24.0%	13.8	60.3%
6	7577	546	7.21%	3422	45.2%	1882	24.9%	13	61.8%
7–39	15,909	1200	7.54%	8510	53.5%	4128	26.0%	11.7	65.0–100%

**Table 4 geriatrics-08-00100-t004:** All admissions. Gender, age, comorbidities, numbers of medications, PIMs, PPOs, Charlson Index, and resource intensity weighting (RIW) and correlations with readmission or mortality within 6 months of discharge (data for 285,201 admissions).

**Risk Factor.**	**Readmission within 6 Months**			**Mortality within 6 Months**		
**Unadjusted Odds Ratios and 95% Confidence Intervals**						
	**OR**	**95%CI**	** *p* **	**OR**	**95%CI**	** *p* **
Gender	1.12	1.10–1.14	<0.001	1.10	1.08–1.12	<0.001
Age at admission	1.01	1.01–1.01	<0.001	1.07	1.07–1.07	<0.001
Number of comorbidities	1.01	1.01–1.01	<0.001	1.19	1.18–1.19	<0.001
Post admission number of medications	1.05	1.04–1.05	<0.001	1.19	1.18–1.19	<0.001
Total STOPP PIMs	1.17	1.17–1.17	<0.001	1.04	1.03–1.04	<0.001
START omissions not corrected	1.58	1.55–1.61	<0.001	2.16	2.11–2.21	<0.001
START omissions correctly prescribed	1.50	1.47–1.52	<0.001	0.99	0.97–1.01	0.189
AGS Beers PIMs	1.11	1.1–1.12	<0.001	1.09	1.09–1.10	<0.001
Charlson Index	1.09	1.08–1.09	<0.001	1.43	1.42–1.43	<0.001
**ORs and 95%CIs adjusted for age at admission, gender (male) and comorbidities**						
**Risk factor**	**OR**	**95%CI**	** *p* **	**OR**	**95%CI**	** *p* **
Post admission number of medications	1.12	1.12–1.12	<0.001	1.04	1.04–1.05	<0.001
Total STOPP PIMs	1.16	1.15–1.16	<0.001	0.99	0.96–1.00	<0.001
START omissions not corrected	1.39	1.35–1.42	<0.001	1.56	1.50–1.63	<0.001
START omissions correctly prescribed	1.26	1.23–1.30	<0.001	0.51	0.49–0.53	<0.001
AGS Beers PIMs	1.11	1.11–1.11	<0.001	1.08	1.07–1.08	<0.001

Data for the numbers of admissions are derived from three files LAB_UC-2022, PIN-US_2022, and DAD_UC_2022 with DAD-UC-2022 containing 295,236 admission records. PIM = potentially inappropriate medication; PPO = potential medication omission.

## Data Availability

Alberta Health Services will not permit researchers to share data outside AHS.

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
