# Peer review of "Multi-Year Retrospective Analysis of Mortality and Readmissions Correlated with STOPP/START and the American Geriatric Society Beers Criteria Applied to Calgary Hospital Admissions"

_geriatrics, 2023, doi:10.3390/geriatrics8050100_

Round 1
Reviewer 1 Report (Previous Reviewer 2)
Abstract
The content of the abstract appears appropriate. Please revise some typing errors (lines 22, 25, 28, 29, 30, 32).
Introduction
The introduction seems appropriate. Please uniform the number of significant digits (lines 52, 57, 58, 64, 68, 69, 70, 73, 74, 75, 89, 91) and how numbers are written (numbers or letters) (lines 50, 55). Please also revise line 48.
Methods
Please revise some typing errors and missing punctuation (lines 105, 115, 120, 131, 145, 153).
Results
Please revise some typing errors in Table 1 and lines 203, 242, 257, 263, 265, 311, 350, 355. Please also revise the number of significant digits in lines 208, 209, 210, 215, 216, 219, Table 3, line 364.
Please explain the abbreviations in Figure 6.
Several typing errors and the punctuation should be corrected.
Author Response
Responses to Reviewer 1. Many thanks for your detailed and very helpful suggestions. |
|
Abstract The content of the abstract appears appropriate. Please revise some typing errors (lines 22, 25, 28, 29, 30, 32). |
All typing errors have been corrected. |
Introduction The introduction seems appropriate. Please uniform the number of significant digits (lines 52, 57, 58, 64, 68, 69, 70, 73, 74, 75, 89, 91) and how numbers are written (numbers or letters) (lines 50, 55). Please also revise line 48. |
All numbers have been corrected to the number of appropriate digits. Numbers are now consistently written as letters e.g. sixth and not as a number e.g. 6th. |
Methods Please revise some typing errors and missing punctuation (lines 105, 115, 120, 131, 145, 153). |
All typing errors and missing punctuation have been corrected. |
Results Please revise some typing errors in Table 1 and lines 203, 242, 257, 263, 265, 311, 350, 355. Please also revise the number of significant digits in lines 208, 209, 210, 215, 216, 219, Table 3, line 364 |
Typing errors in Table 1 and lines 203, 242, 257, 263, 265, 311, 350, 355 have been corrected. The number of significant digits in lines 208, 209, 210, 215, 216, 219, Table 3, line 364 have been corrected. For those who died in hospital the percentages were carried to two decimal points because small differences in such an important outcome would be the subject of close analysis and decisions. |
Please explain the abbreviations in Figure 6. |
The abbreviations have been explained. |
Comments on the Quality of English Language Several typing errors and the punctuation should be corrected. |
These have been corrected. |
Reviewer 2 Report (New Reviewer)
I commend your work on addressing the important topic of the contribution of PIMs and PPOs contributing to hospital readmission and mortality in older adults. I agree with your decision to review this topic using the STOPP/START and Beers criteria, the two most common system used to evaluate appropriate medication usage in older adults.
Overall, the manuscript is incredibly dense and covers a large number of topics… from the history of readmissions/morality in hospitalized older adults to inappropriate medication usage in this group to the potential impact of supportive care such as communication between pharmacist and PCPs in reducing readmission/mortality. At times, the amount of information presented can feel a bit overwhelming for the reader. However, given the scope of your undertaking (reviewing hospital admissions/readmissions and mortality at four acute care Calgary hospitals from 2013-2021), this scope seems warranted. In addition, given your many references, readers could consult the literature on the individual topics if they wished.
The study was well designed and this is adequately described. The healthcare system in Canada enabled you to review the data on hospitalization and medication for the study population. However, there are limitations to this data set, which you review, including inability to access patient-level data and information regarding ADRs. The results are clearly presented and the supporting figures/tables do an excellent job of visually presenting the data.
While section 4.9, destination after discharge, is interesting I feel it is not critical for the manuscript and less closely tied to the purpose of the study. If needed, it could be cut to decrease length and overall complexity.
Your comments regarding generalizability of the study are accurate. This will need to be considered by readers considering the implications of this study for more ethnically diverse populations of older adults.
Overall, the quality of English language in this manuscript is excellent. I noted a few grammatical issues where comma usage was incorrect, but this was very minor.
Author Response
Responses to Reviewer 2. |
|
I commend your work on addressing the important topic of the contribution of PIMs and PPOs contributing to hospital readmission and mortality in older adults. I agree with your decision to review this topic using the STOPP/START and Beers criteria, the two most common system used to evaluate appropriate medication usage in older adults. |
Thank you for your very kind and encouraging comments. |
Overall, the manuscript is incredibly dense and covers a large number of topics… from the history of readmissions/morality in hospitalized older adults to inappropriate medication usage in this group to the potential impact of supportive care such as communication between pharmacist and PCPs in reducing readmission/mortality. At times, the amount of information presented can feel a bit overwhelming for the reader. However, given the scope of your undertaking (reviewing hospital admissions/readmissions and mortality at four acute care Calgary hospitals from 2013-2021), this scope seems warranted. In addition, given your many references, readers could consult the literature on the individual topics if they wished. |
The study is the largest and most complex in the in the literature. We felt duty bound to represent as thoroughly as possible the experiences of the ~ 295,000 seniors who were admitted to the hospitals. |
The study was well designed and this is adequately described. The healthcare system in Canada enabled you to review the data on hospitalization and medication for the study population. However, there are limitations to this data set, which you review, including inability to access patient-level data and information regarding ADRs. The results are clearly presented and the supporting figures/tables do an excellent job of visually presenting the data. |
Thank you for your comments. The hospital staff enter an enormous amount of data many times daily for each patient in the database for the Alberta Health Services commented “it is very difficult for researchers to extract data from the database.” And they are right! |
While section 4.9, destination after discharge, is interesting I feel it is not critical for the manuscript and less closely tied to the purpose of the study. If needed, it could be cut to decrease length and overall complexity. |
We were very keen to reflect the experiences of the patients. During my professorial duties in addition to my hospital and clinic duties I volunteered to be the medical superintendent of four different nursing homes at different times, the largest being for 200 patients. I was always concerned that patients were discharged from hospital to home in as many cases as possible – they are generally happier and less likely to contact infections there. |
Your comments regarding generalizability of the study are accurate. This will need to be considered by readers considering the implications of this study for more ethnically diverse populations of older adults. |
Yes, I have published a series of articles on the P450 system and ethnic variability in alleles. |
Comments on the Quality of English Language Overall, the quality of English language in this manuscript is excellent. I noted a few grammatical issues where comma usage was incorrect, but this was very minor. |
The grammatical errors have been corrected. |
This manuscript is a resubmission of an earlier submission. The following is a list of the peer review reports and author responses from that submission.
Round 1
Reviewer 1 Report
Please make the title shorter and more presis. Focus on the research phenomenon and the design.
The abstract is very long. It should be made shorter and with 300 words.
The introduction on pages 2-3 has no citation to support your writing and sentences.
On page 6, after all descriptions, before the goal, write about the gap that encouraged you to do this study.
For the methods section, please be more detailed and you are suggested to use the equator checklist to organise it. Please fill it out and attach it to the article as supplementary file in resubmission.
The Strengthening the Reporting of Observational Studies in Epidemiology (STROBE) Statement: guidelines for reporting observational studies | EQUATOR Network (equator-network.org)
Nothing.
Reviewer 2 Report
Title
Title could be improved by not including numbers.
Abstract
The content of the abstract is appropriate for the topics of the manuscript but it requires the correction of some typing errors and other minor revision, such as:
- Line 17: form should be from
- Line 20: please revise the sentence as cardiac, pulmonary, CNS, renal, gastro-intestinal are not diagnoses
- Line 22: please remove ; after partial
- Line 30: I suggest to add “many START medication are cardiac or related to other illnesses”
- Line 31: point missing
- Line 48: I suggest to modify the sentence “to prescribe the correct PPOs…”
Formatting should also be standardized.
Introduction
This introduction does not sound like an introduction as a background on potentially inappropriate prescriptions/omissions is missing. Lines 65-91 are the aims of the study and should be moved to the Aim section.
Lines 92-96 are a method and it should be rewritten to provide an overview on existing studies on readmission. The same applies to sections 1.1-1.5, which as they are written seem more results than introduction and should be rewritten to provide an overview on existing studies on readmission.
Please also revise typing errors and punctuation.
Study goals
Should be integrated with lines 65-91.
Methods
The method for measuring OR is missing, please add.
Results
- Section 4.1: Lines 293-298 should be put before Table 1
- Table 1: it is not clear if Total Visit 285201 is the total number of visit or is the total number of visits only for males. Overall age could be changed to Median age and the empty row could be removed. IQR should be a range? In the last rows, columns should be correctly aligned
- Table 2 should be cited in the text. Also please add percentage for greater clarity
- Line 314 and following: I suggest records instead of rows
- Lines 315-320 should be in the discussion section and not in the results
- Table 3: please explain all abbreviations after the table (i.e., Avg). Also, author should declare in the method section how relative risk was measured
- Section 4.4: percentage after the numbers could be useful
- Line 342: Misc? Please explain
- Line 355: COPD is an acronym and it should be explained
- Figure 1 and following: colours should be explained in the caption(s)
- Table 6: authors should specify in the method section how these OR were calculated
- Lines 449-453 should be in the discussion section
- Line 485: authors should explain why this intervention (“not the patients' usual care”) is considered a retrospective study
- Lines 498-502 should be in the discussion section
- Figure 5: 0% could be removed for clarity
Discussion
The discussion section should include a discussion of the main results and not only strengths and limitations of the study.
- Line 577: authors do not mention logistic regression in the method section. Please revise the method section.
Conclusions
- Lines 602-620 do not add anything to the result section
Please revise the discussion and the conclusion sections by including comments on the results in the discussion and by shortening the conclusion to the main arguments.
Several typing errors and the punctuation should be corrected.
Round 2
Reviewer 1 Report
Nothing more.